# Reasons for loss to follow-up (LTFU) of pulmonary TB (PTB) patients: A qualitative study among *Saharia*, a particularly vulnerable tribal group of Madhya Pradesh, India

**Prashant Mishra[1], Ravendra K. Sharma[2], Rajiv Yadav[1], V. G. Rao[1], Samridhi Nigam[1], Mercy Aparna Lingala[1], Jyothi Bhat** [1]*

1 ICMR-National Institute of Research in Tribal Health (NIRTH), Jabalpur, India, 2 ICMR-National Institute of Medical Statistics (NIMS), New Delhi, India

* drjyothibhat@rediffmail.com

## Abstract

### Background

Loss to follow-up (LTFU) among pulmonary tuberculosis (PTB) patients is a significant challenge for TB control. However, there is a dearth of information about the factors leading to LTFU among marginalized communities. This study highlights the factors associated with LTFU in Saharia, a tribe of Madhya Pradesh having high tuberculosis (TB) prevalence.

### Methods

A qualitative study was carried out during January-April 2020 among twenty-two pulmonary TB patients, recorded as LTFU in NIKSHAY, with ten treatment supporters and ten patient's family members. Semi-structured personal interview tools were used to collect the information on the history of anti-tuberculosis treatment, adverse drug events (ADE), social cognitive, behaviors, myths, and misbeliefs. The interviews were transcribed and thematically analysed to examine underlying themes.

### Results

The study explored various social, behavioral factors leading to loss to follow-up among PTB patients. Drug side effects, alcoholism, social stigma, lack of awareness of the seriousness of the diseases and poor counseling are the main barriers to treatment adherence in this community.

### Conclusions

The study highlights the need to address the issues related to LTFU during TB treatment. The enhanced efforts of treatment supporters, health staff, and family & community persons must motivate and support the patients.

**Data Availability Statement:** The manuscript is based on qualitative study. Hence the raw data has names of the participants. So, a formal

administrative and ethical clearances is required for sharing raw data/information. The Institutional ethical committee's Member Secretary, Dr. Tapas Chakma, Scientist 'G' may be contacted for data access requests (tapas_chakma@rediffmail.com).

**Funding:** The study is financially supported by the Government of Madhya Pradesh (Budget 2210/2017-18/877 dated_27/01/18). However, the funding agency had no role in the study design, data collection, analyses, interpretation of results, reports & manuscript writing and submission to the journal.

**Competing interests:** The authors have declared that no competing interests exist.

# Introduction

India is the highest tuberculosis (TB) burden country globally, accounting for about a quarter of the world's TB cases [1]. Though the Revised National Tuberculosis Control Programme (RNTCP), now National Tuberculosis Elimination Programme (NTEP), has been successfully implemented in India for more than two decades, tuberculosis is still a major challenge in the country, especially in rural and remotes areas. Directly observed treatment short course (DOTs) is the backbone of India's TB elimination program, with treatment compliance as an effective strategy [2]. This treatment strategy is adopted by NTEP for almost three decades in the country and is used for both drug-susceptible (DS) and drug-resistant (DR) TB patients under programmatic conditions using multiple anti-tubercular drugs. The standardized treatment regimens for both drug-susceptible TB and drug-resistant TB are used under NTEP. The regimens comprising of 4 drugs for 6–9 months (isoniazid, rifampicin, pyrazinamide, ethambutol) are used for drug susceptible TB and comprising of 6 drugs for (kanamycin, ofloxacin, ethionamide, pyrazinamide, ethambutol, and cycloserine) are used for drug-resistant TB for 24 months. The programme has now introduced a shorter MDR TB regimen and all oral longer MDR TB regimen with new drugs like Bedaquiline and Delamanid can be modified according to Drug susceptibility test (DST) results. The patients are required to attend the hospital at the end of the intensive as well as continuation phase. During the visits, necessary investigations including sputum examination are performed and the health condition of the patients is reviewed. All these anti-tubercular drugs are known for their various side effects. Though treatment providers are expected to support the patients in treatment continuation, there is no community-based treatment support available to patients in this tribal community. Poor compliance to tuberculosis treatment is a major obstacle for TB elimination programme as it can increase the risk of drug resistance and prolong infectiousness and may result in unfavourable outcomes such as treatment failure, death, and relapse, thus posing a public health threat [3, 4]. This is particularly relevant for the tribal population, an underprivileged group of the society usually residing in remote rural areas, thus having poor access to the health delivery system.

Loss to follow up (LTFU) has been defined by the World Health Organization (WHO) as "patients whose treatment was interrupted for two consecutive months or more" [5]. LTFU results from several factors such as access to health services, socioeconomic status, literacy level, and beliefs and practices prevalent in society [6, 7]. Several other studies conducted earlier in different settings have also identified many factors responsible for LTFU: illiteracy, poverty, long duration of treatment, many medicines, access to health services, work-related issues, smoking & alcoholism beliefs & practices, etc. [3, 8, 9]. However, limited studies on the factors responsible for LTFU among TB patients are available in India, especially from resource-constrained settings, including remote tribal areas.

Madhya Pradesh (MP) state in central India alone accounts for 14.7% of the country's total tribal population [10] and tribes exhibited poor health seeking behaviour and access to health services [11]. The Saharia is one of the Particularly Vulnerable Tribal Groups (PVTG) in the state of Madhya Pradesh, central India. Their habitats are located outside the main village called as Saharana.

They are characterized by a primitive economy, socio-economic development and poor nutritional status. They migrate from one place to another in search of livelihoods [12, 13]. Tuberculosis is a key public health problem among them. The studies carried out among this tribe reported a very high prevalence of pulmonary tuberculosis (PTB), 1270 in 1991–92 [14], 1518 in 2008 [15], and 3294 in 2015 [16] per 100,000 in selected *Saharia* areas. A population-based study carried out during 2013–2015 also showed a high prevalence of PTB (3003 per

100,000) among them [17]. Drug-resistant TB is an emerging public health problem in the Saharia tribe [18]. Bhat et al., however, reported that the rates for drug-resistant TB were not different from the national average [19]. The study also reported that most patients with MDR TB in this community were previously treated patients. There is, however, no information available on LTFU among drug resistant tuberculosis patients and the factors associated with LTFU of TB patients amongst the Saharia tribal population. We, therefore, conducted a qualitative study to obtain a better understanding of TB patients, family members, and treatment provider-related factors for LTFU among the Saharia tribe of Madhya Pradesh, central India.

## Methods

### Study setting and population

The study was carried out as a sub-study of the ongoing integrated TB control project (ITCP) in the Saharia population residing in seven districts of the Gwalior and Chambal divisions of Madhya Pradesh state in central India. The districts are linked to drug-resistant TB center (DRTBC) Gwalior, Madhya Pradesh. Drug sensitive (DS) TB patients are provided medicine from the nearest TB DOT center by DOTs providers. The drug-resistant (DR) TB patients are treated at the district TB center (DTC) initially during the pre-treatment evaluation period and later receive treatment from the nearest TB DOT center.

### Study design and sampling

A qualitative study was carried out to explore the reasons for loss to follow-up (LTFU) among Saharia pulmonary TB (PTB) patients. Thirty-Eight PTB (DS and DR) patients (31 males and seven females) were recorded as LTFU in NIKSHAY from the list of TB patients registered for treatment under the integrated TB control project January-October 2019. The survey team visited all LTFU patients from January-April 2020. Among these, five LTFU patients died, three migrated to other districts, and eight were on re-treatment. Rest all twenty-two LTFU patients (19 males and three females) were successfully interviewed and these include thirteen drug-resistant and nine drug-sensitive pulmonary TB patients. Ten treatment supporters and family members of ten patients were also interviewed. Purposive sampling was conducted to ensure maximum variation in responses of treatment supporters and family members.

### Data collection tool and technique

All interviews were carried out using pre-designed open-ended in-depth interview guides developed separately for TB patients (S1 Text: LTFU Patient interview guide), family members (S2 Text: Family member interview guide), and treatment supporters (S3 Text: DOTS provider interview guide). These guides were prepared in consultation with the help of clinicians, DOTS providers in the field, and subject experts and social scientists at ICMR-NIRTH, Jabalpur. The semi-structured interview guides include structured questions on their background information, like age, sex, education status, occupation and past/family history of anti-TB treatment (ATT), and open-ended questions related to adverse drug events (ADE), social cognitive, behaviors, myths and misbelieve that determine adherence and factors associated with loss to follow-up.

   All in-depth interviews were conducted in local regional language (a dialect of *Hindi*) by project scientists (PS) and district coordinators (DC's) of the project who possessed a post-graduate degree either in Life Science, or Public Health, or Social Science disciplines and are trained in TB diagnosis and treatment guidelines. The researchers have been working with the study population for more than one year and are familiar with the local dialect of the Saharia

tribe. But, none of the researchers belonged to Saharia tribal community. The in-depth interviews, each lasting on an average of forty minutes, were carried out during the researchers' routine field monitoring visits. All study participants were interviewed in isolation at a place convenient to them and interview sessions were also audio-recorded with the prior permission of participants.

## Data analysis

The audio recordings of participant's responses were initially transcribed in the Hindi language, later translated into the English language by project research scientists. The transcripts were read by two researchers in the field and codes were developed. All these transcripts were also read and audio recordings were heard by a senior social scientist at ICMR-NIRTH, Jabalpur to ensure that the experiences of the participants were accurately captured and reflected in the inductive codes. The transcripts were coded and categorized into different themes and sub-themes relevant to the study objective. An analysis system based on a review of the literature and a preliminary evaluation of the qualitative data acquired in the context of the research question was developed to examine the relationship between themes and understand the various reasons for loss to follow-up, data were reviewed to identify the perspectives of different types of respondents that have been selected to ensure maximum variation to understand similarities and differences between two groups and among the various respondents for triangulating the findings or differences. Some direct verbatim quotes that showed important responses under each theme are also presented in the manuscript.

## Ethical considerations

This was part of the main study approved by the Institutional Ethics Committee (IEC) with reference no. NIRTH/IEC/2273/2016. Participants who were willing to participate and provided written informed consents were enrolled as study participants.

## Results

### Socio-demographic profile and living conditions of participants

The study participants comprised 22 TB patients reported as LTFU patients in NIKSHAY, aged 25 to 70, including 19 male and three female patients, all belonging to the Saharia tribal community. Among the LTFU patients, thirteen patients were drug-resistant—2 H-Mono resistant, 9 Rifampicin resistant or RR-MDR, and 2 were XDR (Table 1).

The majority (18/22) of the patients were underweight (BMI <18.5) and previously treated (19/22). Most of the respondents are illiterate (18/22) and work as manual labours at stone quarries, agriculture fields, road/house constructions, or other development projects, which is their primary livelihood source. They also migrate to nearby districts and states in search of work/jobs and remain outside of their villages for a substantial period of the year. Most of their houses are Kutcha houses (walls and/or roofs made of mud/stone) or huts (thatched walls and/or ceiling) with a single room and mostly without ventilation. The same room is also used for cooking. The primary fuel used for cooking is wood or crop residuals producing a lot of smoke, one of the significant risk factors of pulmonary tuberculosis (Table 2).

The in-depth interviews of participants identified several factors leading to LTFU in the Saharia community. The themes developed during the analysis of in-depth interviews and important findings are summarized below -

Table 1. Background information of patients (N = 22).

| Factors | Sex | Age | Previous Treatment history | DR Status | Resistant Type |
|---|---|---|---|---|---|
| Patient No.1 | M | 30 | Yes | Yes | RR |
| Patient No.2 | F | 38 | Yes | Yes | H-Mono |
| Patient No.3 | M | 45 | Yes | No | |
| Patient No.4 | M | 36 | Yes | No | |
| Patient No.5 | M | 25 | Yes | No | |
| Patient No.6 | M | 45 | Yes | No | |
| Patient No.7 | M | 60 | Yes | No | |
| Patient No.8 | M | 45 | Yes | Yes | RR |
| Patient No.9 | M | 50 | Yes | Yes | RR |
| Patient No.10 | M | 34 | New | Yes | RR |
| Patient No.11 | M | 70 | Yes | Yes | H-Mono |
| Patient No.12 | M | 32 | Yes | No | |
| Patient No.13 | M | 45 | New | No | |
| Patient No.14 | M | 60 | Yes | Yes | RR |
| Patient No.15 | M | 40 | Yes | No | |
| Patient No.16 | M | 50 | Yes | Yes | RR |
| Patient No.17 | M | 35 | Yes | Yes | RR |
| Patient No.18 | M | 32 | Yes | Yes | RR |
| Patient No.19 | F | 30 | New | No | |
| Patient No.20 | M | 42 | Yes | Yes | XDR |
| Patient No.21 | F | 65 | Yes | Yes | RR |
| Patient No.22 | M | 45 | Yes | Yes | XDR |

## Adverse drug effects

The adverse drug effects (ADE) of medications have been a significant barrier for treatment adherence in the study population. The majority of patients (eighteen out of twenty-two patients) reported some kind of adverse drug effects, commonly reported ADE of TB medication were vomiting, severe headache, vertigo, stomach ache, nausea, and haemoptysis. The adverse drug effects were reported by both drug-sensitive and drug-resistant TB patients.

A 30 years old DR male patient reported–*"When I used to take medicines, I used to get blood out of my mouth, I used to vomit and had trouble in breathing! So, I quit the medicine."* [Patient 1]. Similarly, a 45 years old DS male reported *"Whenever I took medicine, there was severe pain in the stomach, a burning sensation in the urine, the medicine used to heat a lot! So, I quit the medicine."* [Patient 3].

Treatment providers and patient's family members also reported drug side effects as a reason for treatment discontinuation and LTFU among patients.

A 42 years old family member (patient's wife) told—*"He was not taking medicines because he was unhappy with medicine, he had a burning sensation in his stomach and nervousness"* [Family member 4,] whereas a 25 years old male treatment supporter reported—*"Patient s do not take medicine properly because medicine causes heat in the stomach, and vomiting and diarrhea occur, so they quit medicine after few days, and they approach local Gunia (traditional healer) or private practitioner)"* [Treatment supporter 6].

However, more patients perceived adverse drug effects as the important reason for discontinuing treatment compared to the treatment providers and family members.

**Table 2. Common themes that emerged during in-depth interviews of LTFU TB patients.**

| S. No. | Themes | Sub-Themes | Drug-Sensitive (N = 09) | Drug-resistant (N = 13) |
|---|---|---|---|---|
| I | Living condition (Household related) | a) Head of family | Yes -07 | Yes- 09 |
| | | b) House type | Hut/Kutcha- 02 | Hut/Kutcha-09 |
| | | c) Mode of cooking | *Chulla*– 08 | *Chulla* -13 |
| | | d) Place of cooking | Indoor– 04 | Indoor– 09 |
| | | e) Ventilation | No– 03 | No– 09 |
| II. | Personal and lifestyle related | a) Education | Illiterate -06, Primary -03 | Illiterate -11, Primary -02 |
| | | b) Alcohol | Yes– 04 | Yes– 05 |
| | | c) Smoking | Yes -04 | Yes -07 |
| | | d) Myths and misbelief | Yes– 02 | Yes-06 |
| III. | Socio-economic factors related | a) Social stigma and discrimination | Social Stigma -1 Discrimination– 01 | Social Stigma -04 Discrimination -01 |
| | | b) Lack of family and social support | Family support -01 Social support– 01 | Family support -02 Social support– 03 |
| | | c) Unemployment and financial constraints | Unemployment -04 Finical constraints—07 | Unemployment -10 Finical constraints– 11 |
| | | d) Migration for wages | Yes– 04 | Yes -04 |
| IV. | Service provider related | a) Behaviour of treatment supporter | Satisfactory-09 | Satisfactory-13 |
| | | b) Poor counseling | Yes– 04 | Yes– 03 |
| V. | Medications related | a) Adverse drug and treatment effects | Dizziness—02, Fatigue-04, Vertigo-03, Vomiting-02, Nausea-01, Stomach ache-03 | Dizziness -06, Fatigue-04, Vertigo-07, Vomiting-07, Nausea-04, Stomach ache-03 |
| | | b) Long duration regimen | Yes– 0 | Yes– 08 |
| | | c) High pill burden | Yes– 0 | Yes– 08 |

## High pill burden and drug quality

In addition to adverse drug effects, a high pill burden is also an important reason for LTFU in TB patients in this community. The daily regimen of 9–11 pills with injection made it difficult for some patients to complete treatment mainly among drug-resistant patients. A drug-resistant patient reported–

*"Though I had to take 8–10 pills every day, I was taking these pills but I was also asked to take injections. As I was already weak, I used to get dizziness after injection and go into slumber"* [Patient 8, Male, 45 years].

However, few patients also perceive that apart from the quantity, the poor quality of drugs is also a reason for loss to follow-ups and the drug's poor effects.

A 34 years old drug-resistant patient complained of the quality of the drugs and reported- *"The pills were not of good quality, I had to take 8 pills, it was troublesome to take these medicines regularly"* [Patient 10].

The high pill burden and fear of injection among TB patients is also recognized as a reason for LTFU by treatment provides. As one treatment supporter reported *"Patients give up medicine as they have to take many pills, and due to the fear of injections* [Treatment supporter 08, Male, 25 years].

## Work-related and financial constraint

Work-associated problems are also identified as one of the reasons for the discontinuation of the treatment among patients. Fourteen patients reported that they had to quit their job to

start the treatment. Some patients mentioned difficulties in carrying out their works after taking medicines. These Saharia TB patients are economically poor and they mainly work as daily wage agriculture or a manual labourer.

A 35 years old male DR- patient reported *"I used to go to stone quarry work. Since I got sick and I am unable to go to work. Earlier, I used to earn Rs. 50–100 daily, now that too has gone"* [Patients 17,]. While few others mentioned financial constraints in continuing their treatment, and because of their poverty they have to work despite their sickness. A 42 years old male XDR patient reported-

> "*As I had taken payment in advance and was working away from my home, I could not go to my house, so I quit medicine*" [Patient 20].

When they are too weak to continue their work, their family members substitute them so that their family could sustain them. As a 30 years old DS male patient reported *"I used to earn Rs 250–300 daily, but I can no longer go to work due to my illness. So, now, my wife goes to work"* [Patient 03]. A family member also reported-

> "*Since he is sick, I and my son go to earn outside, I want him to get well soon and start earning so that we do not have to worry. But he has stopped taking medicine due to feeling dizziness after medication. We are troubled by working outside*" [Family member 4, Female, 42 years].

## Alcohol abuse and smoking

Habits like alcohol use and tobacco consumption are prevalent among the Saharia community. During the interviews, many patients reported alcohol consumption and smoking as a factor for not adhering to treatment. These were also written by treatment supporters (DOTS providers) and the patient's family members. However, after getting the disease, many of them also stopped smoking and alcohol consumption.

A 40 years old male DS patient mentioned *"Earlier I used to smoke beedi and drink liquor, but now I have given up since I got sick"* [Patient 15]. Another 50 years old DR male reported *"Earlier, I used to drink liquor, but now I have not consumed liquor from last 2–3 years. Now, I smoke only 2–4 Beedi per day, I have been smoking since I was 10–12 years old."* [Patient 9]. But few patients do not stop alcohol consumption and some prefer to stop taking medicine over alcohol. A 35 years old female family member (patient's wife) reported—*"My husband is not taking medicine because he drinks liquor. I have explained to him many times about the bad effects of liquor, but he does not listen to me. He is still suffering from disease, he coughs all the day"* [Family member, 2].

However, few patients also consume or continue to consume alcohol to suppress the symptoms of disease and side effects of the medicine. A 45 years old DS-patient reported *"Yes, I drink liquor, I drink to suppress cough and I take approximately 200 ml at a time"* [Patient 3].

A treatment supporter also mentioned, *"Many patients drink liquor and smoke (Beedis) during the treatment, due to which the medicine cause more heat/burning sensation (adhik garmi karti hai) and they give up the medicine instead of stopping drinking and smoking"* [Treatment supporter 4, Male, 29 Year].

## Stigma and family support

Social stigma, discouragement, and low-income family support are also reported as significant causes and barriers leading to discontinuation of TB treatment. The fears of discrimination by their family members adversely affect the activities promoting adherence and TB treatment

outcomes. During the interview sessions, few TB patients reported that they did not want health care workers to visit their homes for counseling. They did not want to visit their local or block and district level DOT center due to fear of disclosure of their disease status.

A 38 years old DR female patient reported—*"When my TB treatment started, I was very ashamed to tell my neighbor and the villagers. I do not know why, but there was always a fear; I did not tell anyone my treatment was for what"* [Patient 2]. A 30 years old male DR patient also mentioned *"When I got sick, no one from my family was willing to accompany me to the hospital. Only my wife was willing to come, but I did not take her to the hospital because she had never visited a hospital, she does not know anything about hospitals"* [Patient 1]. Few family members also informed that they prefer to keep the patient outside of the house to avoid infection to other family members. A 35 years old family member (patient's wife) told—*"We live separately, though he is my husband. He lives in a hut outside the house, because it is a communicable disease"* [Family member 2].

## Myths and misbelieve

Lack of awareness about the seriousness of the disease and its treatment also influenced treatment adherence adversely. Many patients who did not complete their treatment had myths and misbeliefs associated with the TB disease and treatment. Some patients did not trust diagnosis and continue to believe that they do not have TB disease. So, they were reluctant to start treatment and stop taking medicine after a few days/ weeks of treatment initiation. A 45 years old male DS patient reported–

"*I do not have any disease, so why should I take medicine.*" [Patient 13].

However, some also believed that taking medicine or anti-TB treatment will further deteriorate their health.

A 50 years old male DR patient reported, *"I would have not been able to sit again if I had taken these pills and would have died. People in my family also used to get upset and used to say why I should take such pills"* [Patient 9].

A few of them also perceived that different patients get different anti-TB medicines. One who received good medicines he/she completes treatment, while others who don't receive good medicines had a side effect.

A 45 years old male DS patient revealed—"*My younger brother had the same disease (TB). He also took medicine but the medicines he was taking were different. My medicine caused stomach aches and a burning sensation in the urine. My medicine was not good, he got better medicines*" [Patient 6]. Whereas, some patients also expressed peer pressure to stop medicines. Sixty years old male DR-patient reported–

"*I was taking medicine for a long time and when I started getting blood in the cough, my neighbors and villagers asked me to stop taking too many medicines. So, I quit medicine*" [Patient 14].

During interviews, it was also observed that patients, care providers, and family members also correlate misfortunes with TB disease or its treatment. A 38 years old DR female patient who lost her child during the treatment reported–

"*When I was taking medicines, my son died. I was very much disheartened and left the medicines. When my son is no more, then for whom should I take medicines?*" [Patient 2].

### Challenges and barriers (representative's quotations)

Treatment supporters and patient's family members were also interviewed to identify the challenges and barriers leading to LTFU. Some of them mentioned that long duration of treatment, alcoholism, and smoking are important causes of LTFU. As reported by one treatment supporter–

"*As far as I have experienced, the medicines have adverse effects (garmi karti hai) and the duration of treatment is prolonged. So, patients take medicine for one or two months and then they quit medicine*" [Treatment supporter 1, Male, 20 years].

Another treatment supporter reported-

"*Many patients drink liquor daily and get drunk and do not take medicines regularly. Subsequently, they stop medicine* [Treatment supporter 4, Male, 29 years].

Some patients stop the medicine after taking one-two months when symptoms get subsided, thinking that they are disease-free and thus do not take a full course of treatment. A treatment supporter reported–

"*When patients take proper medicine regularly, they start to feel better within a month, and they start thinking that they have recovered from the disease and quit the medicines*" [Treatment supporter 2, Male, 20 years]. Similarly, another treatment supporter revealed -

"*After taking medicines for 1–2 months, patient's health improves and he thinks that he is cured, and start avoiding medicines, and finally stop the medicine*" [Treatment supporter 3, Male, 35 years].

The important quotations of the drug-sensitive and drug-resistant study participants are given below (Table 3)-

## Discussion

This is the first reported qualitative study that provides insight into the causes of LTFU of TB patients among the Saharia- a PVTG in Madhya Pradesh with a very high TB burden. India's National Tuberculosis Elimination Programme reported 4% LTFU for the TB patients notified in 2018 [20]. A recent study from central India reported 8.6% LTFU among TB patients registered under NTEP [21]. Our study among the Saharia tribe found a high proportion of unfavorable treatment outcomes including a higher rate of post-treatment mortality [22] and LTFU (15.3%) among Saharia TB patients (unpublished data). The low socio-economic status, frequent migration to other areas, under-nutrition, poor access to health facilities, and inadequate reach of the health system due to remotely located Saharia habitats might be contributing to higher LTFU in this community. The study focused on in-depth analyses of narrative data from patients, treatment supporters, and patient's family members.

We found that patients' judgment of the severity of their fatigue, dizziness, vomiting, and high pill burdens also led to LTFU among the Saharia. The adverse drug events (ADE) and high pill burden of TB regimen promote discontinuation of treatment or patients shift to private TB treatment, especially among DR patients. The poor awareness of the drug's possible side effects and the absence of counseling before initiating medications contribute to the LTFU. Without proper counseling, patients drift out of the system and discontinue the treatment. Several other studies also reported similar findings among TB patients [3, 21, 23–26]. In

**Table 3. Theme-wise important quotations of the drug-sensitive and drug-resistant study participants.**

| Drug Sensitive participants | Drug Resistant participants |
|---|---|
| **Adverse drug effects** | |
| *"Whenever I took medicine, there was severe pain in the stomach, a burning sensation in the urine, the medicine used to heat a lot! So, I quit the medicine."* [Patient 3, Male, 45 years]. <br> *"I used to vomit after taking medicine, So, I quit this medicine."* [Patient 05, Male, 25 years]. | *"When I used to take medicines, I used to get blood out of my mouth, I used to vomit and had trouble in breathing! So, I quit the medicine."* [Patient 1, Male, 30 years]. <br> *"When I used to take pills, I had severe pain in the stomach, vomiting, and diarrhea, there was a burning sensation in the stomach, and also felt dizziness and nervousness. So, I quit this medicine."* [Patient 10, Male, 34 years]. |
| **High Pill burden and drug quality** | |
| | *"Though I had to take 8–10 pills every day, I was taking these pills but I was also asked to take injections. As I was already weak, I used to get dizziness after injection and go into slumber"* [Patient 8, Male, 45 years]. <br> *"The pills were not of good quality, I had to take 8 pills, it was troublesome to take these medicines regularly"* [Patient 10, Male, 34 years]. |
| **Work-related and financial constraint** | |
| *"I used to go to daily wages work. Since I got sick and I am unable to go to work. Earlier, I used to earn Rs. 200–250 daily, now that too is gone otherwise, I would have earned 6–7 thousand in a month"* [Patients 04, Male, 35 years]. <br> *"I used to earn Rs. 250–300 daily; I can no longer go to work due to illness. Now wife goes to work"* [Patients 03, Male, 35 years]. | *"I used to go to stone quarry work. Since I got sick and I am unable to go to work. Earlier, I used to earn Rs. 50–100 daily, now that too is gone"* [Patients 17, Male, 35 years]. <br> *"When I was taking medicine, I used to get dizziness and I was afraid that I wouldn't get my wages"* [Patient 16, Male, 50 years]. <br> *"As I have taken payment in advance and working away from my home, I cannot go to my house, so I quit medicine"* [Patient 20, Male, 42 years]. |
| **Alcohol abuse and smoking** | |
| *"Yes, I drink liquor, I drink to suppress cough and I take approximately 200 ml at a time."* [Patient 3, Male, 45 years]. <br> *"Earlier I used to smoke beedi and drink liquor, but now I have given up since I got sick"* [Patient 15, Male, 40 years]. | *"Earlier, I used to drink liquor, but now I have not consumed liquor for the last 2–3 years. Now, I smoke only 2–4 Beedi per day, I have been smoking since I was 10–12 years old."* [Patient 9, Male, 50 years]. <br> *"Earlier I used to smoke Ganja (Marijuana) and I smoked for 4–5 years, but now I have given up since last 3–4 months"* [Patient 10, Male, 34 years]. |
| **Stigma and family support** | |
| *"When my TB treatment started, my wife started living separately from me, I have to live in a hut (Jhopdi) outside the home and my food is also prepared separately"* [Patient 04, Male, 34 years]. | *"When my TB treatment started, I was very ashamed to tell my neighbor and the villagers. I do not know why, but there was always a fear; I did not tell anyone my treatment was for what"* [Patient 2, Female, 38 years]. <br> *"When I got sick, no one from my family was willing to accompany me to the hospital. Only my wife was willing to come, but I did not take her to the hospital because she had never visited a hospital, she does not know anything"* [Patient 1, Male, 30 years]. |
| **Myths and misbelieves** | |
| *"I do not have any disease, so why should I take medicine."* [Patient 13, Male, 45 years]. <br> *"My younger brother had the same disease (TB). He also took medicine, but the medicines he was taking were different. My medicine caused stomach aches and a burning sensation in the urine. My medicine was not good, he got better medicines"* [Patient 6, Male, 45 Year]. | *"I would have not been able to sit again if I had taken these pills and would have died. People in my family also used to get upset and used to say why I should take such pills"* [Patient 9, Male, 50 years]. <br> *"I was taking medicine for a long time and when I started getting blood in the cough, my neighbors and villagers asked me to stop taking too much medicine. So, I quit medicine"* [Patient 14, Male, 60 years]. <br> *"When I was taking medicines, my son died; I was very much disheartened and left the medicines. When my son is no more, then for whom should I take medicines?"* [Patient 2, Female, 38 years]. |

the community setting, particularly in remote tribal areas, when ADE occurs, patients need advice and counseling but they hardly find any health staff nearby to support them. So, they are compelled to approach local practitioners and traditional healers readily available in the hamlet/locality. Our findings show that the patients were not fully aware of the possible side effects of drugs and were not aware of how to cope with these adverse effects, particularly in the intensive phase of treatment. Other studies have also reported similar findings in different regions [23, 24, 27]. The possible reason for higher ADE among the Saharia tribe could be the high prevalence of undernutrition among them. Patients with poor nutritional status are reported to have a higher risk of hepatotoxicity which is one of the important ADE due to anti-TB drugs, thus contributing to LTFU [28, 29]. The BMI rates are lower in this tribe, and adequate nutrition deprivations likely to lead to severe reactions like vomiting and nausea, thus promoting discontinuation of TB treatment among patients. The pre-treatment counseling and active surveillance of patients' adverse reactions by the health staff are crucial in improving treatment compliance.

Our work shows that alcohol consumption during treatment is a significant barrier to treatment compliance in the Saharia tribe. The findings are consistent with previous investigations. Alcoholism has been reported as a significant factor of patient non-compliance or adverse treatment outcomes among tuberculosis patients receiving DOTS treatment [26, 30–33]. Because of these findings, the history of alcoholism before starting treatment would help identify potential defaulters. We also found smoking leading to treatment discontinuation in the present study. Similar findings were observed among TB patients receiving standard TB regimen in different settings [31, 34]. The Saharia tribe is one of Madhya Pradesh's marginalized communities, and they depend on daily work for their livelihood. Most of the LTFU patients are either primary earners or are in economic active ages, work primarily on the daily wages. The financial constraints and personal reasons related to work, i.e., fear of losing a job and migration within the country for the job, are key factors responsible for losing follow-up and not completing the treatment. Several other studies also reported work and financial constraints as the possible reasons for treatment discontinuation [23, 30, 35–37].

The Saharia tribe is a particularly vulnerable tribal group of Madhya Pradesh with extensive illiteracy. Our study also showed that most of the study participants are illiterate. Thus, incorrect knowledge or myths, and misconceptions about TB disease and its medications and treatments are common among patients and family members. The low literacy could also be why their incorrect knowledge about TB disease has been reported in several studies [26, 38, 39]. This highlights the importance of appropriate communication techniques to be adopted by the health workers while working in such communities with high illiteracy. We found the cultural interpretation for the drugs as 'hot', 'heat generating' in this tribal community, which is similar to a finding among Vietnamese refugees [40]. We also found many myths and misbeliefs prevalent among the patients and family members affecting treatment compliance. Several other studies also reported similar findings among TB patients [41, 42]. In the present study, some patients denied TB diagnosis and were trying to hide the disease. Other studies also reported similar perceptions of TB patients in different settings [43, 44]. Traditional healers are usually a part of the tribal society, and the community has complete faith in them. So, they are consulted first for any ailment in addition to being easily approachable and locally available. We also found this faith in 'Gunia' (traditional healers) among the Saharia tribe. A study in South Africa also reported a preference for traditional healers among TB patients [23].

The complications and the factors associated with LTFU reinforced the view that only medication's free distribution is not enough for treatment adherence or the cure. The stigma associated with tuberculosis also plays an important role in treatment compliance. We found that stigma has a strong influence on family and community members. A similar influence of

stigma is also reported in various studies conducted in different countries [23, 24, 37]. Social support, support from family, friends, neighborhoods, and community are extremely important for TB treatment adherence. Our study revealed that lack of social and family support to TB patients obliges them to stop or interrupt the treatment. Thus, family and community support are essential to promote early diagnosis and complete treatment. Several other studies also reported social/family support influence, including financial and emotional support, on treatment adherence [23, 24, 37, 45].

Overall, the study revealed that multiple factors are responsible for LTFU among the Saharia tribe and highlighted the importance of better awareness, motivational counseling of patients and family members, and community involvement in TB diagnosis and treatment. The study also established the need for financial support and nutritional supplements for TB patients. The enhanced efforts of treatment supporters, health staff, and family & community persons are required to motivate and support the patients. Further research to devise appropriate strategies to improve treatment compliance should be undertaken as part of TB control in high endemic areas/communities.

## Limitations

The study is based on an in-depth interview of 22 Saharia tribe TB patients and provides substantial information about the factors leading to LTFU among the *Saharia* PVTG of Madhya Pradesh, but one needs to be cautious while generalizing the findings.

The major strength of this study is that it provides information from patients, service providers, and family member's perspectives and also from both drug-susceptible (DS) and drug-resistant (DR) TB patients.

## Supporting information

**S1 Text. LTFU patient interview guide.**
(DOC)

**S2 Text. Family member interview guide.**
(DOC)

**S3 Text. DOTS provider interview guide.**
(DOC)

## Acknowledgments

The authors are thankful to the Director ICMR-NIRTH, Jabalpur for his support and encouragement throughout the study period. Sincere thanks are due to the district coordinators of the project for carrying out the study in remote tribal villages and all the field staff for their help during the data collection. We are also thankful to the study participants who spared their time and provided valuable information during the study. The institute publication committee approved the manuscript (ICMR-NIRTH/PSC/46/2020).

## Author Contributions

**Conceptualization:** Prashant Mishra, Ravendra K. Sharma, Jyothi Bhat.

**Formal analysis:** Ravendra K. Sharma.

**Investigation:** Prashant Mishra, Samridhi Nigam.

**Methodology:** Ravendra K. Sharma, V. G. Rao, Mercy Aparna Lingala, Jyothi Bhat.

**Project administration:** Prashant Mishra, Ravendra K. Sharma, Rajiv Yadav, Samridhi Nigam, Mercy Aparna Lingala, Jyothi Bhat.

**Supervision:** Rajiv Yadav, V. G. Rao, Jyothi Bhat.

**Validation:** Prashant Mishra.

**Writing – original draft:** Prashant Mishra, Ravendra K. Sharma, V. G. Rao, Jyothi Bhat.

**Writing – review & editing:** Prashant Mishra, Ravendra K. Sharma, Rajiv Yadav, V. G. Rao, Jyothi Bhat.

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
