## [Decision Letter · Decision Letter 0]

5 May 2021

PONE-D-20-21794

Factors leading to loss to follow-up (LTFU) of pulmonary TB (PTB) patients: A Study among Saharia a particularly vulnerable tribal group of Madhya Pradesh, India

PLOS ONE

Dear Dr. Bhat,

Thank you for submitting your manuscript to PLOS ONE. After careful consideration, we feel that it has merit but does not fully meet PLOS ONE’s publication criteria as it currently stands. Therefore, we invite you to submit a revised version of the manuscript that addresses the points raised during the review process.

Please submit your revised manuscript. If you will need significantly more time to complete your revisions, please reply to this message or contact the journal office at plosone@plos.org. Please include the following items when submitting your revised manuscript:

We look forward to receiving your revised manuscript.

Kind regards,

Frederick Quinn

Academic Editor

PLOS ONE

Journal Requirements:

2. Please include the study type in the title (i.e. qualitative study).

3. Please include additional information regarding the interview guide used in the study and ensure that you have provided sufficient details that others could replicate the analyses. For instance, if you developed the guide as part of this study and it is not under a copyright more restrictive than CC-BY, please include a copy, in both the original language and English, as Supporting Information.

4. Please discuss any land access permits or Saharia tribal permissions obtained for this research.

Reviewers' comments:

Reviewer's Responses to Questions

**Comments to the Author**

1. Is the manuscript technically sound, and do the data support the conclusions?

Reviewer #1: Yes

Reviewer #2: Yes

Reviewer #3: Yes

Reviewer #4: Partly

2. Has the statistical analysis been performed appropriately and rigorously? 

Reviewer #1: N/A

Reviewer #2: Yes

Reviewer #3: N/A

Reviewer #4: N/A

3. Have the authors made all data underlying the findings in their manuscript fully available?

Reviewer #1: Yes

Reviewer #2: Yes

Reviewer #3: Yes

Reviewer #4: No

4. Is the manuscript presented in an intelligible fashion and written in standard English?

Reviewer #1: Yes

Reviewer #2: Yes

Reviewer #3: Yes

Reviewer #4: Yes

5. Review Comments to the Author

Reviewer #1: Comments for the paper entitled “Factors leading to loss to follow up (LTFU) of pulmonary TB (PTB) patients: A study among Saharia a particularly vulnerable tribal group of Madhya Pradesh, India”

Overall comments

This manuscript focused on an important topic to identify the Factors leading to loss to follow up of Saharia a particularly vulnerable tribal pulmonary TB patients. Overall this research address important issue and well written paper I recommended for publication with minor revisions and worth emphasising the following points.

Specific comments

Abstract: Need to add details on what is the design of the study, who collected data, what are the information collected.

Introduction

The authors should strengthen the introduction by providing more information on Rationale and novelty of the study in the introduction as they quote this is the first qualitative study undertaken among the Saharia Tribe. It was mentioned that several other studies conducted earlier in different settings have also identified many factors responsible for LTFU such as illiteracy, poverty, long duration of treatment, large number of medicines, access to health services, work related issues, smoking & alcoholism, beliefs and practices etc. However, limited studies on the factors responsible for LTFU among TB patients are available in India. Why, the authors expecting any different finding from this area, if so whether they identified any new factors responsible for loss to follow-up this population.

Methods

• Drug sensitive (DS) TB patients are provided medicine through DOT providers and linked to the nearest TB DOT center, whereas, drug resistant (DR) TB patients are treated at district TB center (DTC) initially during pre-treatment evaluation period and later receive treatment from the nearest TB DOT center. But the analysis is combined with DS and DR. Why this was done separately, because the treatment itself different.

• Status of loss to follow-up patients need explanation. Reasons for five deaths, is it due to loss to follow-up or any other reasons.

• Authors included all the sample, why it called as a purposive sample – if so on what criteria?

Results

• Age of the patient was 25-70 is wide range, is it OK

• Table 2 need explanation

• How drug reaction was measured, any definition used or any medical officer defined. Who assesse this problem is related to drugs

Discussion

• What is the loss to follow-up in other areas, is it different from other areas, if so what is the possible reasons to be disused

• Provide reference for higher ADE with under nutrition.

• How counselling will help in reducing ADE

• The findings are consistent with previous investigations is it in Saharia tribe or other, if so then there is no difference between Saharia and other population.

• The recommendations are not relevant to reduction of ADE , how better awareness, motivational counseling of patients and family members, and involvement of community in TB diagnosis and treatment will help to reduce ADE.

• “Social and culturally acceptable interventions along with patient centric health facilities are essential to improve treatment compliance in high endemic areas/community” this is not from this study findings, need to be modified.

Reviewer #2: Manuscript needs major corrections

The manuscript needs vast grammar corrections

All the highlighted sentences are needed to be revised before to submit

Results parts are to be revised and give briefly

Reviewer #3: General Comments:

An important paper that deals with a marginalized population with high incidence of TB. They are some typographical and grammatical errors that should be corrected.

• The title however could be adjusted to better reflect the content of the study. Instead of factors associated with (which implies a quantitative study), perhaps: Reasons for …….

• The introduction should better describe the TB treatment program, regimens and duration for MDR TB and DS TB. How often patients visit the hospitals, any community-based adherence treatment support available to patients? any social risk mitigation available to the patients?

• More information on the TB burden among the Saharia tribe (both DS TB and MDR TB) and on magnitude of loss to follow-up among the Saharia tribe in India ((both DS TB and MDR TB). Are most patients with MDR TB in this region previously treated patients?

Methods:

The methods have been described however more detailed description of certain elements of the study would improve its strength e.g.

• The study population is mixed (patients with MDRTB and DS TB). Identification of patient quotes should indicate if which type of TB patient had. It is possible that some findings e.g. use of many pills and injectables may be specific to patients with MDR TB.

• More detailed description of study design including methodological orientation of the study (theoretical framework), duration of interviews, a note on data saturation.

• Data analysis: how many coders analyzed the data, analysis approach (were themes identified in advance or derived from the data), software used to analyze the data-if any.

• Study findings: a discussion of minor themes or diverse cases

• Service provider related reasons mentioned in Table 2 are not represented in the results.

• Figure 1 combines the pathways of care with the reasons for loss to follow-up making it difficult to follow. The casual pathways in the figure are also difficult to understand. The diagram suggests a casual pathway between ADE and family support or ADE and poor counselling.

Discussion:

• “Our findings revealed that approaching the health facility for diagnosis and treatment initiation (step-III) and enrolment/follow-up (step-IV) are very crucial for treatment adherence among TB patients (Fig-I)”. This statement is not supported by the results.

• Study Limitations: Qualitative studies do not usually discuss their strengths and weaknesses in the way that quantitative studies do.

Reviewer #4: This study among the tribal population is a worthy addition to the knowledge in this area. However, this appears to be a lost opportunity as some very important factors were not explored in detail to observe the link and the direction of relationship, leading to a weak conclusion without appropriate recommendations coming from the study. other comments are as follows.

1. Factors for LTFU would be different for the drug-sensitive and drug resistant TB. They should be presented in a stratified manner. Combining them would lead to loss of information for both type of cases.

2. The type of analysis, whether content, framework or grounded theory, is not mentioned.

3. The topic is well researched and most factors are already known. As authors approach the analysis with a knowledge of these factors, they may actually end up doing a deductive coding instead of indicative coding as stated.

4. In the methods, it is mentioned as semi-structured whereas results mention the interviews as in-depth. Both are not same.

5. Whether the interviewers were trained in conduct of qualitative interviews?

6. Responses must have been transcribed in the local language and then translated into English.

7. The web diagram does not seem to be derived from the responses of participants rather from the investigators’ prior knowledge of the relationships as there is no linking of the factors in the results section. The links only shows up in the diagram.

8. Steps 1 and 2 in the web diagram are not relevant as there are no factors acting at those levels.

9. In the discussion section, the web diagram is not discussed. Each factor has been dealt independently as in quantitative studies.

10. Overall language correction is needed.

6. PLOS authors have the option to publish the peer review history of their article (what does this mean?). If published, this will include your full peer review and any attached files.

Reviewer #1: No

Reviewer #2: No

Reviewer #3: No

Reviewer #4: **Yes: **Sonali Sarkar

---

## [Author Response · Author response to Decision Letter 0]

13 Jul 2021

We have addressed all the comments from editor and reviewers.

Point wise reply file is uploaded.

---

## [Decision Letter · Decision Letter 1]

15 Sep 2021

PONE-D-20-21794R1Reasons for loss to follow-up (LTFU) of pulmonary TB (PTB) patients: A qualitative study among Saharia, a particularly vulnerable tribal group of Madhya Pradesh, IndiaPLOS ONE

Dear Dr. Bhat,

Thank you for submitting your manuscript to PLOS ONE. After careful consideration, we feel that it has merit but does not fully meet PLOS ONE’s publication criteria as it currently stands. Therefore, we invite you to submit a revised version of the manuscript that addresses the points raised during the review process.

Please submit your revised manuscript. If you will need significantly more time than this to complete your revisions, please reply to this message or contact the journal office at plosone@plos.org. Please include the following items when submitting your revised manuscript:A rebuttal letter that responds to each point raised by the academic editor and reviewer(s). You should upload this letter as a separate file labeled 'Response to Reviewers'.A marked-up copy of your manuscript that highlights changes made to the original version. You should upload this as a separate file labeled 'Revised Manuscript with Track Changes'.An unmarked version of your revised paper without tracked changes. You should upload this as a separate file labeled 'Manuscript'.If applicable, we recommend that you deposit your laboratory protocols in protocols.io to enhance the reproducibility of your results. Protocols.io assigns your protocol its own identifier (DOI) so that it can be cited independently in the future. For instructions see: https://journals.plos.org/plosone/s/submission-guidelines#loc-laboratory-protocols. Additionally, PLOS ONE offers an option for publishing peer-reviewed Lab Protocol articles, which describe protocols hosted on protocols.io. Read more information on sharing protocols at https://plos.org/protocols?utm_medium=editorial-email&utm_source=authorletters&utm_campaign=protocols.

We look forward to receiving your revised manuscript.

Kind regards,

Frederick Quinn

Academic Editor

PLOS ONE

Journal Requirements:

Reviewers' comments:

Reviewer's Responses to Questions

**Comments to the Author**

1. If the authors have adequately addressed your comments raised in a previous round of review and you feel that this manuscript is now acceptable for publication, you may indicate that here to bypass the “Comments to the Author” section, enter your conflict of interest statement in the “Confidential to Editor” section, and submit your "Accept" recommendation.

Reviewer #2: All comments have been addressed

Reviewer #3: All comments have been addressed

2. Is the manuscript technically sound, and do the data support the conclusions?

Reviewer #2: Yes

Reviewer #3: Yes

3. Has the statistical analysis been performed appropriately and rigorously? 

Reviewer #2: Yes

Reviewer #3: N/A

4. Have the authors made all data underlying the findings in their manuscript fully available?

Reviewer #2: Yes

Reviewer #3: No

5. Is the manuscript presented in an intelligible fashion and written in standard English?

Reviewer #2: Yes

Reviewer #3: Yes

6. Review Comments to the Author

Reviewer #2: Minor corrections required

Minor grammar mistakes are to be corrected and unwanted , has to be removed where it does not required

Reviewer #3: I thank the authors for taking the time to respond to initial comments raised. I have a few minor additional comments to bring to the authors' attention

General comments:

The authors have made significant effort to improve the manuscript.

Other comments

a) Under Study setting: Please indicated the lost to follow-up rates in this region from your project data

b) Under Study Design: Please state the theoretical framework underpinning this study

Line 66: Write DOTs in full the first time it is used

Line 135: The sentence construction needs improvement

Line 155 : add the length of time taken by each indepth interview

Line 225: LTUF is misspelt

Line 422: states "our study showed that most respondents were illiterate". Respondents literacy levels are not presented anywhere in the study results

Table 1: Summarize baseline characteristics instead of listing them for each participant

Table 3 is not necessary as it is a repetition of what is presented in the text.

7. PLOS authors have the option to publish the peer review history of their article (what does this mean?). If published, this will include your full peer review and any attached files.

Reviewer #2: **Yes: **Muthaiah Muthuraj

Reviewer #3: No

---

## [Author Response · Author response to Decision Letter 1]

19 Oct 2021

Thank you for your comments. 

We have addressed all the queries. Hope you find it useful.

---

## [Decision Letter · Decision Letter 2]

8 Nov 2021

PONE-D-20-21794R2Reasons for loss to follow-up (LTFU) of pulmonary TB (PTB) patients: A qualitative study among Saharia, a particularly vulnerable tribal group of Madhya Pradesh, IndiaPLOS ONE

Dear Dr. Bhat,

Thank you for submitting your manuscript to PLOS ONE. After careful consideration, we feel that it has merit but does not fully meet PLOS ONE’s publication criteria as it currently stands. Therefore, we invite you to submit a revised version of the manuscript that addresses the points raised during the review process.

Please submit your revised manuscript Dec 23 2021 11:59PM. If you will need significantly more time to complete your revisions, please reply to this message or contact the journal office at plosone@plos.org. Please include the following items when submitting your revised manuscript:A rebuttal letter that responds to each point raised by the academic editor and reviewer(s). You should upload this letter as a separate file labeled 'Response to Reviewers'.A marked-up copy of your manuscript that highlights changes made to the original version. You should upload this as a separate file labeled 'Revised Manuscript with Track Changes'.An unmarked version of your revised paper without tracked changes. You should upload this as a separate file labeled 'Manuscript'.

We look forward to receiving your revised manuscript.

Kind regards,

Frederick Quinn

Academic Editor

PLOS ONE

Journal Requirements:

Reviewers' comments:

Reviewer's Responses to Questions

**Comments to the Author**

1. If the authors have adequately addressed your comments raised in a previous round of review and you feel that this manuscript is now acceptable for publication, you may indicate that here to bypass the “Comments to the Author” section, enter your conflict of interest statement in the “Confidential to Editor” section, and submit your "Accept" recommendation.

Reviewer #2: All comments have been addressed

Reviewer #3: All comments have been addressed

2. Is the manuscript technically sound, and do the data support the conclusions?

Reviewer #2: Yes

Reviewer #3: Yes

3. Has the statistical analysis been performed appropriately and rigorously? 

Reviewer #2: Yes

Reviewer #3: N/A

4. Have the authors made all data underlying the findings in their manuscript fully available?

Reviewer #2: Yes

Reviewer #3: No

5. Is the manuscript presented in an intelligible fashion and written in standard English?

Reviewer #2: Yes

Reviewer #3: Yes

6. Review Comments to the Author

Reviewer #2: All the pointe are carefully addressed by author

however manuscripts needs Minor corrections required

Reviewer #3: We thank the authors for taking the time to response to reviewers' comments and improve the manuscript. All my previous comments have been addressed.

7. PLOS authors have the option to publish the peer review history of their article (what does this mean?). If published, this will include your full peer review and any attached files.

Reviewer #2: **Yes: **Muthuraj Muthaiah

Reviewer #3: No

---

## [Author Response · Author response to Decision Letter 2]

15 Nov 2021

Thank you for all the suggestions.

Suggested corrections have been made in the manuscript.

---

## [Decision Letter · Decision Letter 3]

29 Nov 2021

Reasons for loss to follow-up (LTFU) of pulmonary TB (PTB) patients: A qualitative study among Saharia, a particularly vulnerable tribal group of Madhya Pradesh, India

PONE-D-20-21794R3

Dear Dr. Bhat,

We’re pleased to inform you that your manuscript has been judged scientifically suitable for publication and will be formally accepted for publication once it meets all outstanding technical requirements.

Kind regards,

Frederick Quinn

Academic Editor

PLOS ONE

Additional Editor Comments (optional):

Reviewers' comments:

Reviewer's Responses to Questions

**Comments to the Author**

1. If the authors have adequately addressed your comments raised in a previous round of review and you feel that this manuscript is now acceptable for publication, you may indicate that here to bypass the “Comments to the Author” section, enter your conflict of interest statement in the “Confidential to Editor” section, and submit your "Accept" recommendation.

Reviewer #2: All comments have been addressed

Reviewer #3: All comments have been addressed

2. Is the manuscript technically sound, and do the data support the conclusions?

Reviewer #2: Yes

Reviewer #3: Yes

3. Has the statistical analysis been performed appropriately and rigorously? 

Reviewer #2: Yes

Reviewer #3: Yes

4. Have the authors made all data underlying the findings in their manuscript fully available?

Reviewer #2: Yes

Reviewer #3: Yes

5. Is the manuscript presented in an intelligible fashion and written in standard English?

Reviewer #2: Yes

Reviewer #3: Yes

6. Review Comments to the Author

Reviewer #2: (No Response)

Reviewer #3: I thank the authors for taking the time to address all comments provided by the reviewers during the various rounds of review. I have no further comments.

7. PLOS authors have the option to publish the peer review history of their article (what does this mean?). If published, this will include your full peer review and any attached files.

Reviewer #2: **Yes: **Dr.Muthuraj Muthaiah,Government Hospital for Chest Diseases,Puducherry.

Reviewer #3: No

---

## [Editor Report · Acceptance letter]

14 Dec 2021

PONE-D-20-21794R3 

**Reasons for loss to follow-up (LTFU) of pulmonary TB (PTB) patients: A qualitative study among *Saharia*, a particularly vulnerable tribal group of Madhya Pradesh, India**

Dear Dr. Bhat:

I'm pleased to inform you that your manuscript has been deemed suitable for publication in PLOS ONE. Congratulations! Your manuscript is now with our production department. 

Kind regards, 

on behalf of

Dr. Frederick Quinn 

Academic Editor

PLOS ONE